# Mucoadhesive Buccal Film of Estradiol for Hormonal Replacement Therapy: Development and In-Vivo Performance Prediction

**DOI:** 10.3390/pharmaceutics14030542

**Published:** 2022-02-28

**Authors:** Sadikalmahdi Abdella, Franklin Afinjuomo, Yunmei Song, Richard Upton, Sanjay Garg

**Affiliations:** 1Pharmaceutical Innovation and Development (PIDG) Group, Clinical and Health Sciences, University of South Australia, Adelaide, SA 5000, Australia; sadikalmahdi.abdella@mymail.unisa.edu.au (S.A.); franklin.afinjuomo@unisa.edu.au (F.A.); may.song@unisa.edu.au (Y.S.); 2School of Pharmacy, College of Health Sciences, Addis Ababa University, Zambia St., Addis Ababa 1000, Ethiopia; 3Clinical and Health Sciences, University of South Australia, Adelaide, SA 5000, Australia; richard.upton@unisa.edu.au

**Keywords:** estradiol, film, mucoadhesive, patient-centric, menopause, in vivo prediction

## Abstract

The age-related loss of circulating estrogen that occurs during the menopausal transition manifests itself through a variety of symptoms including vasomotor (hot flushes and night sweats), genito-urinary syndrome (vaginal dryness and urinary symptoms), sexual dysfunction, mood, and sleep disturbance that often last longer than a decade. Furthermore, reductions in estrogen level increase the risks of chronic complications such as osteoporosis, cardiovascular disease, and cognitive decline among others, thereby affecting the quality of life of women. Although oral estrogens are the most widely used therapy for menopausal symptoms, they suffer from poor bioavailability, and there are concerns over their safety, creating a significant concern to consumers. Mucoadhesive buccal films are an innovative dosage form that offers several advantages including avoidance of the first-pass metabolism, fast onset of action, and importantly, improved patient acceptance. In the current work, we developed mucoadhesive estradiol film for hormonal replacement therapy using film-forming polymers. Two approaches, namely, co-solvency and nano-emulsion were evaluated to increase solubility and hence incorporate estradiol, a poorly water-soluble drug, into a formulation made from the hydrophilic polymer/s. The films were characterised for their mechanical and physicochemical properties. In-vitro release study showed that about 80% of the drug was released within 6 min from films prepared by the nano-emulsion approach, whereas it took about 10.5 min to get similar drug release from films prepared by the co-solvency approach. The ex-vivo permeation result indicates that about 15% of the drug permeated across the porcine buccal mucosa in the first 10 h from films prepared by the nano-emulsion approach, while permeation across porcine buccal mucosa was only observed at around 24 h from films prepared by the co-solvency method. The nano-emulsion films were evaluated for in vivo performance using a convolution technique using R software. The predicted Cmax and Tmax were found to be 740.74 ng mL^−1^ and 7 min, respectively, which were higher than previously reported in vivo concentration from oral tablets. The results demonstrated that mucoadhesive film of estradiol based on the nano-emulsion approach could be a promising platform for the delivery of estradiol through the buccal mucosa for the treatment of menopausal symptoms.

## 1. Introduction

With the progressive aging of the world population, most women, particularly those living in the developed world, likely spend greater than a third of their life beyond menopause, and by 2030, an estimated 47 million women will be undergoing menopause each year [1]. The age-related loss of circulating estrogens that occurs during the menopausal transition manifests itself through a variety of symptoms including vasomotor (hot flushes and night sweats), genito-urinary syndrome (vaginal dryness and urinary symptoms), sexual dysfunction, mood, and sleep disturbance that often last longer than a decade [2]. Among 75–80% of women who experience menopausal symptoms, almost half of them find the symptoms distressing, while 20–30% have severe symptoms making the management of menopause-related symptoms a quite important issue in women’s health [3]. Besides, reduction in estrogen levels increases risks for osteoporosis, cardiovascular disease, and cognitive decline among others [4]. Eliminating menopausal symptoms and associated potential sequelae remains the primary goal of therapy. Several randomized controlled studies have demonstrated that estrogen represents the most effective treatment for menopausal vasomotor symptoms and related issues including impaired sleep, irritability, and decreased quality of life [5]. Hence, in the absence of contraindications, systemic moderate-dose estrogen-containing hormone therapy is currently the most effective treatment for Vasomotor Symptoms (VMS) and associated complications [6].

Estradiol and conjugated equine estrogens are the most widely used forms of oral estrogens for the treatment of menopausal symptoms. Standard doses of oral estradiol and conjugated equine estrogens are 1.0 mg and 0.625 mg, respectively. Higher and lower doses of these formulations are also available [7]. The conventional “one-size-fits-all” approach to estrogen dosing in postmenopausal women was found to carry a risk of adverse effects. Thus, women should take the lowest dose possible, followed by dose titration until the symptom is controlled, for the shortest time possible and without a progestin if feasible [8].

Estrogen is available in many dosage forms: oral, transdermal, topical gels and lotions, and vaginal rings. A standard dose of systemic estrogen (oral 1 mg/day or transdermal 0.05 mg/day) can effectively lessen hot flashes and night sweats in most women with little difference in their ability to alleviate the symptoms. Nonetheless, oral estrogen was found to increase the levels of sex hormone-binding globulin, triglycerides, and C-reactive protein owing to its extensive first-pass hepatic metabolism, which in turn increases the risk of vascular events including venous thromboembolism (VTE) [9,10] creating a significant concern to consumers. Vaginal gels and rings were considered as an alternative but are associated with poor compliance and are less effective in the control of vasomotor symptoms (hot flushes and night sweats). Transdermal treatment was the safest type of hormone replacement therapy when the risk of venous thromboembolism was assessed, but transdermal use is also limited by erratic absorption with large interindividual and intraindividual variability in apparent bioavailability that ranges from 25% to 225%. Factors such as the barrier property of the stratum corneum, the drug physicochemical property, and skin blood flow have been shown to affect transdermal drug absorption. For instance, estradiol levels with estradiol patches are higher in the evening than in the morning, which might be due to circadian variations in skin blood flow [11]. Consequently, the limitation with vaginal formulation, oral tablets, and transdermal patch warrants exploring other administration routes to improve the safety and delivery of estrogen formulations. 

In recent years, significant interest has been shown in the development of mucoadhesive buccal films using bio-adhesive polymers. They are innovative approaches and considered patient-centric dosage forms with high patient acceptability. Characteristics such as small size, comfort, ease of application, storage, handling, and the possibility of taking them with no or just a little water make them an ideal delivery system for most drugs [12]. The buccal transmucosal administration of drugs is a non-invasive and simple route for systemic administration of drugs. It offers several advantages over other routes of administration, such as the rapid onset of action due to its rich vascularization, bypassing the enzymatic degradation of the gastrointestinal tract, avoiding the first-pass metabolism, and possibly improving bioavailability [13,14]. Furthermore, the oral cavity and buccal mucosa are easy to access, which makes application as well as the removal of a drug simple for the patient or the caregiver [15]. This route is exceptionally beneficial for drugs that undergo extensive hepatic first-pass metabolism including estradiol, which has only about 5% oral bioavailability [16]. Previous studies have indicated that hormones such as estrogen and progesterone, in troche formulations, were readily absorbed via the buccal mucous membrane, and their peak plasma concentrations were comparable to those found normally in young menstruating women [17]. The serum concentration and bioavailability of estradiol have been increased by 10- and 5-fold when administered by buccal route compared to the oral route [18]. 

Estradiol is sparingly soluble in water (3.90 mg/L at 27 °C) and soluble in organic solvents such as ethanol (2.5 mg/mL). Preparing fast dissolving oral films of poorly soluble drugs is challenging as it requires dissolving the drugs in hydrophilic polymers. Limited solubility in water not only affects the incorporation of a poorly soluble drug in hydrophilic films but may also result in poor bioavailability, intrasubject or inter-subject variability, and lack of dose proportionality. Several strategies such as co-solvency [19] solid dispersions [20], complexation with cyclodextrin [21] and lipid-based formulations [22], and nano-emulsion [23] have been reported in the literature for solubilizing drug candidates with low aqueous solubility. Among them, the use of cosolvent (i.e., co-solvency) and nano-emulsion are the most popular approaches for improving the solubility of poorly aqueous soluble drugs in pharmaceutical formulation [24].

The objective of this study was to develop an innovative mucoadhesive estradiol buccal film for hormonal replacement therapy. Co-solvency and nano-emulsion approaches were used to incorporate the estradiol into the polymer solution. To the best of our knowledge to date, there is no report of the formulated mucoadhesive buccal film of estradiol, which emphasized the challenge involved in the design and development. We hypothesized that mucoadhesive buccal films of estradiol would achieve higher plasma drug concentration while avoiding the risk of venous thromboembolism. The films were functionally characterized for release kinetics, in vitro muco-adhesion, physicochemical properties, and permeation through buccal mucosa. The originality of this study lies in a formulation of buccal film of estradiol, which overcomes the limitations of the current oral and transdermal route of administration. 

## 2. Materials and Methods

### 2.1. Materials 

Different film-forming polymers were investigated either alone or in combination. Poly (vinyl alcohol) (15,000 Mw 80–89% hydroxylated), polyvinylpyrrolidone (PVP K-30 PI), citric acid, Tween^®^ 80 (Polyoxyethylene Sorbitan Monooleate), Transcutol P oil, sucralose, and Poly (ethylene glycol) (PEG) 400 were acquired from Sigma-Aldrich Merck (St. Louis, MO, USA); A model drug estradiol USP (E2) and xanthan gum were purchased from PCCA (Matraville, New South Wale, Australia). Hydroxypropyl methylcellulose (METHOCEL™ E50) was donated by Colorcon (Melbourne, Victoria, Australia). All chemicals were of pharmaceutical or analytic grade and used as received without any further modification or purification.

### 2.2. Methods

#### 2.2.1. Solubility Enhancement

Two approaches, co-solvency and nano-emulsion were used to improve the solubility of estradiol in hydrophilic polymers. Films prepared by the two techniques were compared in terms of their characteristics, release profile and permeation across porcine buccal membrane. Transcutol P and deionized water formed oily and aqueous phase of the emulsion. Tween^®^ 80 was included in the formulation as a surfactant in both approaches. Ultrasonication was employed to reduce the particle size [25]. 

#### 2.2.2. Preparation of Simulated Salivary Fluid (SSF)

SSF was prepared as reported by Koland et al. [26] with slight modification. Na_2_HPO_4_(238 mg), KH_2_PO_4_ (19 mg), and NaCl (800 mg) were added one by one to 50 mL of de-ionized water. The volume was made up to 100 mL using deionized water following dissolution of the ingredients. Finally, the pH was adjusted to 6.85 using 0.1 N hydrochloric acid. 

#### 2.2.3. Formulation (Gel and Film) Development

Various grades of HPMC namely Methocel E3, Methocel E5, Methocel E15, and Methocel E50 Premium LV, and different molecular weight PVA were evaluated as film formers. In addition, Kollicoat IR (PVA-PEG copolymer) alone or in combination with other polymers was investigated. Glycerin, propylene glycol, and polyethylene glycol were evaluated as a plasticizer. The films were prepared by the solvent casting method and were evaluated for imperfections, peel-ability without rupturing, surface roughness, and appearance. Films that showed good properties upon preliminary evaluation were further optimized (Table 1). 

To prepare the final optimized film (Table 1 F7 & F8), the following procedure was followed; firstly, PVA was soaked in deionized water heated to 90 °C with constant stirring at 2500 rpm to get a clear solution. PVP and xanthan gum were added to the clear solution at the same conditions and stirred until it formed a clear solution. Then, the temperature was reduced to 60 °C and HPMC was added and stirred at 1500 rpm for 20 min to fully dissolve the HPMC followed by continuous stirring at 1000 rpm at room temperature for 2 h. A solution of estradiol was prepared separately. Two approaches were used, co-solvency and nano-emulsion. In the case of co-solvency, PEG 400, Tween 80, ethanol, and the drug were mixed and stirred to obtain a clear solution. For the nano-emulsion approach, PEG 400, Tween 80, Transcutol, and deionized water were mixed and ultrasonicated using an ultrasonic homogenizer (QSonica Q125, Newtown, CT, USA) for 14 min. Finally, drug solutions, prepared by both approaches, were added separately to the polymer solution previously prepared and stirred at 1000 rpm for 2 h to get a uniform solution. The solution was kept in a refrigerator for 12 h to remove air bubbles, casted on a Teflon (6 mm diameter) and allowed to dry at 40 °C for 12 h in an oven. The film was peeled off the casting surface and cut into rectangular sheets (2.2 cm × 3.2 cm) for further studies. Blank films were prepared similarly except with no drug. The polymers and other ingredients incorporated are frequently used in drug formulation development and non-toxic in nature [27,28,29]. Transcutol P is a non-toxic and biocompatible substance that has been used as the main ingredient in cosmetic and over-the-counter topically applied products [30]. 

#### 2.2.4. Characterization of Estradiol Loaded Mucoadhesive Buccal Films

Tensile strength and film adhesion were measured using a texture analyser (Stable Micro Systems, TA.XTPlus texture analyzer Godalming, Surrey, UK). Films were fixed tight onto a platform with screws for measuring film burst (TA-108S5 platform) and muco-adhesion (TA-90 platform). All measurements were performed in triplicate.

#### 2.2.5. Tensile Strength (TS)

A film was fixed between upper and lower clamps then pulled apart at a rate of 1 mm/s with a contact force of 0.05 N. A tensile force required to break the films was measured. The TS of the film was calculated using Equation (1) [31].
(1)TS (MPa)= Force at break[Fmax]Cross−sectional area[A]
where *Fmax* is the maximum force at break, and *A* is the initial cross-sectional area of the film. Comparison between groups was performed using one-way ANOVA in R software. 

#### 2.2.6. Muco-Adhesion

Cellophane membrane (Thermo Fisher Scientific) was used as a model membrane to evaluate muco-adhesion of the films as described by Kumar et al. [32]. The cellophane was hydrated with artificial saliva for an hour at room temperature before use. Then, the hydrated cellophane was cut into a proper size to fit the muco-adhesion support ring and the probe. The test was started after 15 min of equilibration at 37 °C. The films were dissolved in artificial saliva to form a gel and the gel was then loaded into the rig and the test was started 5 min after loading the gel into the rig. A flat-faced probe (TA-10) moved at the speed of 1 mm/s until it touched the gel in the rig (Figure 1). On contact, 0.2 N force was applied for 60 s and the probe was withdrawn at a speed of 1 mm/s. The force required for the probe to detach from each film was recorded as the adhesion force of the films. Adhesion force was measured in triplicate for each film (Figure 1).

#### 2.2.7. Thickness and Weight Uniformity

Thickness and weight uniformity were determined using digital micrometer and electronic balance. The thickness was measured at 5 separate points of each film, and the average thickness was calculated. The weight uniformity of the film was tested by taking the weight of five films of 3.2 cm × 2.2 cm individually, and the average weight was calculated. Comparison between groups was performed using one-way ANOVA in R software.

#### 2.2.8. Content Uniformity

Drug content uniformity was determined by dissolving the buccal film (3.2 cm × 2.2 cm) in 10 mL of a solution (1:1 water and ethanol) in a shaker rotator adjusted to 37 °C. After the dissolution of the film mesh, the solution was centrifuged at 3000 rpm for 5 min to completely precipitate the polymer blend, and the upper layer of the solution was then filtered through 0.45 μm filter (Millipore^®^) to remove any undissolved polymers. After dilution, drug concentration was assayed using the HPLC model (Shimadzu Corporation, Kyoto, Japan) containing a pump (LC-20ADXR), an autosampler, and a photodiode array (PDA) detector. The stationary phase was a C18 column, Alltech Alltima™ (250 mm × 4.6 mm, 5 μm). HPLC analysis was performed at ambient temperature, with a mobile phase of acetonitrile–water (50:50, *v/v*) delivered at 1 mL min^−1^. A detection wavelength of 210 nm and injection volumes of 10 μL were used throughout the study. The concentrations of estradiol were calculated using a calibration curve, produced by plotting peak areas versus concentration. Stock solutions were prepared in the mobile phase. Each preparation was tested in triplicates. Comparison between groups was performed using one-way ANOVA in R software. 

#### 2.2.9. Folding Endurance

Three randomly selected films were taken to measure the folding endurance. The films were repeatedly folded at the same place until they broke. The number of times a film could be folded at the same place without breaking gave the value of the folding endurance.

#### 2.2.10. Surface pH

The surface pH measurements were performed at room temperature by adding a small drop of deionized water on the top of the films and letting them sit for 30 s before measurement. The pH was noted after bringing the electrode of the pH meter in contact with the solution and allowing equilibration for 30 s. The average of three determinations for each formulation was calculated (*n* = 3).

#### 2.2.11. Droplet Size and Polydispersity Index

Droplet size and PDI of the drug solution prepared by nano-emulsion were determined using a dynamic light scattering technique using a Malvern Zeta sizer (Nano ZS, Malvern Instruments Ltd., Malvern, UK). Samples were diluted 100 times with deionized water to prevent multiple scattering phenomena and were analysed by being placed in a disposable polystyrene cuvette at 25 °C and a scattering angle of 173°.

#### 2.2.12. In Vitro Disintegration

The films (3.2 cm × 2.2 cm) were placed in a glass Petri dish (8.5 cm^2^) containing 10 mL of purified water warmed to 37 °C. The plate was swirled manually every 5 s, until the films were dissolved or disintegrated into the suspension. The time was recorded, and the average of 3 samples was calculated. 

#### 2.2.13. In Vitro Dissolution Test

Films (2.2 cm × 3.2 cm) were placed in a falcon tube containing 10 mL of SSF maintained at 37 ± 0.5 °C and stirred at 50 rpm in a shaking water bath (Julabo SW22, Germany). Aliquots of 1 mL were manually withdrawn at 0, 1.5, 3, 4.5, 6, 7.5, 9-, and 13.5-min time intervals and replaced with an equal volume of fresh simulated salivary medium to keep the sink condition of the dissolution medium. The withdrawn samples were analysed for the drug content using HPLC. All experiments were carried out in triplicate.

#### 2.2.14. Permeation Studies

Permeation of the estradiol across freshly excised (<2 h) porcine buccal mucosa was investigated at 37 °C in Franz diffusion cells (*n* = 3), with a 2.54 cm^2^ diffusion area and a compartment volume of 5.2 mL. The receptor chamber was filled with PBS pH 7.4, and the porcine epithelium was mounted between the acceptor and donor compartments. The estradiol film was cut into the size (2.54 cm^2^) and carefully attached to the porcine mucosa. The film was then wetted by adding 400 µL of SSF. Aliquots (1 mL) were withdrawn from the receptor at different time points and dried with nitrogen gas to remove the solvents. The receptor chamber was immediately replenished with an equal volume of fresh and preheated (37 °C) PBS. At the end of the experiment, the buccal tissue was immersed in 10 mL HPLC mobile phase and sonicated for 1 h, to recover the accumulated amount of drug. Quantification of the API was accomplished by HPLC. The steady-state mass flux (Jss) was determined as the slope of the cumulative mass–time plotted curve (linear fraction). Calculation of the apparent permeability coefficient (Papp) was achieved by utilizing equation Papp = Jss/Cd, where Cd is the initial donor-associated API concentration.

#### 2.2.15. Fourier Transform Infrared Spectroscopy (FTIR)

FTIR spectra of pure drug, physical mixtures, blank film, and drug-loaded film were recorded using an ATR-FTIR Perkin Emler spectrum 400 USA, FTIR instrument. The spectra were obtained at a frequency range of 4000–400 cm^−1^ with a resolution of 4 cm^−1^ and 12 scanning rates. The powders and films were placed on the ATR diamond crystal, and subsequently, a force was applied with the use of the clamp to ensure that the samples made adequate contact with the diamond crystal. Spectra were recorded over a range of 4000 to 400 cm^−1^.

#### 2.2.16. Differential Scanning Calorimetry (DSC)

The thermal behaviour of the pure drug, physical mix, and films was studied using Discovery DSC 2920 from TA Instruments (New Castle, DE, USA) calibrated with an indium standard. Pure drug, physical mix, and the films were accurately weighed (2.0 ± 0.5 mg) and placed in the aluminium pans and heated from room temperature to 200 °C under a nitrogen atmosphere with a heating rate of 10 °C/min.

#### 2.2.17. X-Ray Powder Diffraction (XRD)

Powder XRD is an important tool for the prediction of the crystalline nature of any substance. The XRD pattern of the samples (blank and drug-loaded films, physical mixture, and pure drug) was measured using an Empyrean, Malvern Panalytical XRD equipment (Empyrean XRD Worcestershire, Malvern, UK) coupled with a graphite crystal monochromator with filter radiation of Cu-Kα1 (λ = 1.5406 Å) at 30 kV and 30 mA. The diffraction angle for the analysis (2θ) was from 5° to 50° with a scanning speed set at 1.2°/min to accurately measure the crystallinity of the sample.

#### 2.2.18. Scanning Electron Microscopy (SEM)

SEM imaging of pure estradiol and the films were investigated using Zeiss Merlin Field-Emission Gun with silicon drift detector Energy-Dispersive X-Ray Spectroscopy (SDD EDS) (Jena, Germany). Before the analysis, the samples were freeze-dried and subsequently placed on a double-sided tape. Finally, this was sputter-coated with platinum and then examined using accelerating voltage in the range of 2–5 KV.

#### 2.2.19. Prediction of In Vivo Performance

A convolution technique was used to predict in vivo drug concentrations for the buccal films based on the in vitro drug release data determined above and the pharmacokinetics of intravenous estradiol in women obtained from literature [33]. For convolution, the Unit Input Response (UIR) of the drug is identified which represents the systemic kinetics of the drug (i.e., the in vivo concentrations after an i.v. dose of 1 dose unit). The UIR convolved with a new time-course of extravascular input rate (I) to predict the in vivo concentration time-course for the new extravascular administration process. Convolution is an integration process: If the rate at which the drug comes out from a formulation as a function of time is denoted by I(t)and the UIR is denoted by U(t) which is plasma concentration time relationship then the plasma concentration profile C(t) resulting from the entire dose I(t) is given by.
(2)$$C\left(t\right)=\int_{0}^{t}I\left(x\right)\cdot u\left(t-x\right)\mathrm{dx}$$

Convolution was performed using the convolve function in R software (version 1.2.5042) (Appendix B: R script). The buccal bioavailability was assumed to be 100%, based on the high bioavailability of other lipophilic compounds for this route [34]. 

## 3. Result and Discussion

Estradiol is a commonly used drug for the management of menopausal symptoms and is commercially available in different dosage forms including oral tablets, vaginal rings, and topical gels. Although the oral route is an extensively used method, oral estradiol has been linked to a risk of vascular events such as VTE due to extensive metabolism by the liver. The transdermal route can reduce the risk of vascular effects, but it is associated with erratic absorption that leads to inter- and intraindividual variation in bioavailability [14]. Consequently, there is space to improve the safety and patient acceptability of estradiol formulation. Estradiol is an ideal candidate for buccal drug delivery as this route avoids the first-pass metabolism and is associated with increased patient acceptability. Mucoadhesive buccal films of estradiol were developed and optimized as this formulation is innovative and offers several advantages compared to other routes of drug administration. Hydrophilic polymers were used in this study to obtain fast drug release, and two strategies, namely, co-solvency and nano-emulsion methods were used to increase the solubility of estradiol in hydrophilic polymers. Surfactant, heat, and stirring were also applied to facilitate the complete dissolution of the components. Ethanol was used as a cosolvent because estradiol is soluble in ethanol, and ethanol easily evaporates during drying of the film. Transcutol P was used as an oily phase in nano-emulsion as it is a commonly used oil in pharmaceutical formulations. Additionally, Transcutol P could improve penetration of the drug across the buccal mucosa. Air bubbles that usually result in non-uniform distribution of various film components were removed by keeping the final solution in a refrigerator for 12 h before casting. Plasticizers were used to increase flexibility, improve peelability, and reduce the brittleness of the film.

### 3.1. Physical Appearance of the Films

Before incorporating the drug into the polymer solution, blank films were prepared and visually evaluated. Formulations that did not homogeneously coat the casting surface, form pores, or could not be peeled off, were discarded. The drug was incorporated into a blank formulation that formed a good film. Films made of polymer alone were difficult to peel and brittle. Different plasticizers were, therefore, evaluated during the preliminary experiments. The main purpose of using plasticizers is to provide flexibility and improve peelability. Among the used plasticizers, glycerin and PEG provided the desired effect; however, glycerin made the film surface oily. Hence, PEG was selected for the final formulations. Films based on only PVA as a polymer were transparent, thin, and showed stretch marks upon solvent evaporation (Figure 2; Film 1). Adding HPMC into a PVA formulation resulted in a transparent and relatively thick film (Figure 2; Film 3). The ratio of HPMC, PVA, and PEG was optimized to develop a film with a good property. Films based on Kollicoat were transparent, dry, and difficult to peel from the casting surface (Figure 2; Films 4 and 5). 

Estradiol (80 mg) was incorporated into 50 mL of optimized polymer solution (F6, Table 1 and Figure 2); however, the estradiol separated during drying showed white crystals of estradiol at the surface of a film upon drying. To increase the solubility of the estradiol and to avoid the separation of the drug, the two approaches were employed. A solution containing the drug was prepared separately by using the co-solvency and nano-emulsion method and finally added into the polymer solution. The nano-emulsion resulted in a film with no sign of drug separation, but estradiol separated out from the film in films prepared by co-solvency method. To prevent the drug from precipitating out of the solution, PVP was incorporated. Incorporation of PVP increased the viscosity of the formulation, which might have helped to suspend the drug in the film-forming solution. Consequently, films containing PVP showed no sign of drug separation. Additionally, PVP addition improved the thickness of the film. The formulation was finally optimized by adding xanthan gum to increase muco-adhesion. The final drug-loaded films formed by two approaches had a similar appearance and were found to be colourless, peelable, and homogeneous with no evidence of estradiol separation upon visual inspection. The optimized films (F7 and F8) (Table 1) were characterized (Figure 2). 

### 3.2. Tensile Strength (TS) and % Elongation (PE)

In order to determine the flexibility and elasticity of the buccal films, the TS and % elongation (PE) was calculated. Ideally, buccal films are expected to have higher PE and TS [35]. The TS and PE varied from 0.496 to 0.0.552 (MPa) and 10.5% to 15.1%, respectively (Table 2). No marked differences were observed between films prepared by the two approaches.

### 3.3. Thickness and Weight Uniformity

The thickness and weight uniformity of the prepared films were measured with digital micrometer and electronic balance, respectively. Measurement of buccal film thickness and weight is necessary to determine the accuracy of dose in the film. Films prepared by nano-emulsion were heavier than films prepared by co-solvency method, which could be due to the thickness difference between the films, or the molecular weight of the solvents (Table 3). Ethanol evaporates upon drying, while oil used in nano-emulsion will be retained. The data suggest that films prepared by both approaches possess adequate folding endurance (>300) that could help resist breaking or tearing during application and storage (Appendix A). 

It is evident from Table 3 that the prepared buccal films were thin (0.13–0.16 mm) and suitable for buccal application without the potential for causing any discomfort to the patient. Further, the films possessed uniform thickness, demonstrating no significant difference between batches and film-preparation approaches.

### 3.4. Surface pH

To avoid any irritation to the buccal mucosa, it is mandatory to have the pH of the film close to the pH of the saliva. A pH range between 5.5–7.0 is considered to be acceptable [36]. All the fabricated films exhibited pH within the acceptable range. The pH of the optimized films prepared by co-solvency and nano-emulsion approach were 6.51 ± 0.11 and 6.42 ± 0.02 respectively. 

### 3.5. Droplet Size and Polydispersity Index (PDI)

Globule size, and PDI are critical properties of nano-emulsion as they provide information about quality of the formulation. As shown in Figure 3, the average particle size and PDI of the nano formulation were 14.92 nm and 0.487 respectively (Table 4). Low values of PDI for all formulations indicate successful development of a monodispersed system [37].

### 3.6. Drug Content Uniformity

The regression equation of the calibration curve had been used to calculate the drug content (Appendix A). The contents of the drug in the films prepared using co-solvency and nano-emulsion in a 3.2 cm × 2.2 cm film were 1.47 and 1.46 mg, respectively (Table 5). 

### 3.7. Muco-Adhesion Studies

Muco-adhesion studies were carried out to ensure the adhesion of the formulation to the mucosa for a prolonged period at the site of absorption. Films prepared by co-solvency method showed better adhesion compared to films prepared by nano emulsion method. The maximum force of detachment was 0.44 ± 0.005 N for films prepared by co-solvency and 0.34 ± 0.02 N for films prepared by nano emulsion. The lower muco-adhesion in nano emulsion could be due to the presence of oil in the formulation [38]. 

### 3.8. In Vitro Disintegration

The disintegration time of films prepared by the co-solvency and nano emulsion was determined using petri dish method. The films (3.2 cm × 2.2 cm) were placed in a glass Petri dish (8.5 cm^2^) filled with 10 mL of simulated saliva fluid at 37 °C. The disintegration time of the films was noted. The in vitro disintegration time of the films prepared by co-solvency and nano-emulsion approaches was 28.3 ± 2 s and 27 ± 2 s respectively. This could be attributed to the high hydrophilicity of the polymers and other excipients. This result may be an indicator of a fast dissolution of the film and rapid release of the entrapped drug during buccal administration [39]. Fast disintegrating film should disintegrate within 30 s [12].

### 3.9. In Vitro Drug Release

The in vitro drug release studies showed that films prepared by both approaches showed rapid release of estradiol. The films prepared by the nano-emulsion approach released the drug faster. At 6 min, about 80% of the drug was released from films prepared by nano-emulsion approach, whereas it took around 10 min to obtain comparable release from films prepared by the co-solvency method. It took around 7.5 min to get complete release of the drug from films prepared by the nano-emulsion approach, whereas complete release was achieved at 13 min for films prepared by the cosolvency method. The fast release of estradiol from films prepared by nano-emulsion could be attributed to the drug delivery system (Figure 4). 

### 3.10. Mathematical Modeling of Drug Release Profiles

To study the mechanism of estradiol release from the films, various mathematical models (Table 6) were applied to the drug release data obtained from in vitro release studies in the simulated salivary medium. The goodness of fit of the experimental release profiles was evaluated using three common statistical criteria in combination: the adjusted R2, the Root Mean Square Error (RMSE), and the Akaike Information Criterion (AIC).

The experimental release data of estradiol from films prepared by the nano-emulsion approach were best fitted by the Hopefenberg and Korsmeyer–Peppas models, respectively. On the other hand, the release of estradiol from films prepared by co-solvency was fitted by the Weibull model followed by Hopfenberg model (Figure 5). The β value of the Weibull model is the shape parameter that describes the release curve as either sigmoidal (S-shaped) with an upward bend followed by a turning point (when β > 1) or exponential (when β = 1), or parabolic with a steeper initial slope and rest consistent with the exponential (when β < 1). Hopfenberg is an empirical mathematical erosion model of the system; it is assumed that the rate of drug release from the erodible system is proportional to the surface area of the device that is allowed to change with time. All mass transfer processes involved in controlling drug release are assumed to add up to a single zero-order process (characterized by a rate constant, k0) confined to the surface area of the system. This zero-order process can correspond to a single physical or chemical phenomenon, but it can also result from the superposition of several processes, such as dissolution, swelling, and polymer chain cleavage. A good example for systems is Hoffenberg’s model, which can be applied to surface-eroding polymer matrices where a zero-order surface detachment of the drug is the rate-limiting release step. 

### 3.11. Permeation Studies

Ex Vivo permeation studies were performed to establish the absorption kinetics of the drug across the buccal epithelium to the systemic circulation [40]. The primary goal of permeation study is to predict the permeability of drugs in humans, and the method also helps to evaluate potential penetration enhancers to improve buccal transport [41]. 

Permeation studies were conducted using porcine buccal mucosa due to morphological similarity with human epithelium. Permeation profiles of films prepared by nano-emulsion-based technology are illustrated in Figure 6. Permeation of estradiol from films prepared by the co-solvency method was not observed at all time points except at 24 h. The amount permeated at 24 h was only 5.8% highlighting insignificant permeation of estradiol from films prepared by the co-solvency approach. This finding that illustrates increasing solubility does not guarantee permeation through membranes. Transcutol P, which is commonly used as a penetration enhancer, and a small droplet size of nano-emulsion-based film could have helped in the fast release of estradiol from the nano-emulsion formulation. 

Estradiol permeation across porcine buccal mucosa from films prepared by nano-emulsion based technology showed a relatively linear relationship with time. The steady-state flux (Js) of o-estradiol through the mucosa (calculated from the linear portion of the data between 360 and 480 min) and the permeability coefficient (Kp) were determined to be 0.121 ± 0.018 μg cm^−2^ h^−1^ and 3.0 × 10^−5^ cm h^−1^, respectively. Although both approaches helped in increasing dissolution of the drug in the formulation, hence, forming a uniform film with no evidence of crystallinity, estradiol film prepared by the nano-emulsion method has shown better permeation across buccal mucosa (Figure 6. The enhanced permeation of estradiol from nano-emulsion could be related to the fact that nanoparticles increase the diffusion rate of the drug across the mucus layer, adhere to the buccal mucosa prolonging the buccal residence, and increase the contact time with the mucosa [42]. 

### 3.12. Fourier Transform Infrared (FTIR) Analysis

FTIR study was performed to evaluate interactions between the film matrix and the drug. Spectra of estradiol, blank, and drug-loaded films, as well as the physical mixture of HPMC, PVA, PVP, xanthan gum, citric acid, and sucralose, were evaluated. The spectra of estradiol show strong and characteristic peaks of the drug. Two broad intense peaks at 3393 and 3462 cm^−1^ are due to O–H stretching vibration showing extensive hydrogen bonding in the solid-state. 

As reported in the literature the FTIR of estradiol showed a broad intense O–H stretching peak at 3393 and 3462 cm^−1^. Peaks observed in the range of 1350–1230 cm^−1^ can be attributed to the free hydroxyls and the intermolecular hydrogen bonds. Peaks in the range of 2980–2807 cm^−1^ are related to stretching C–H vibrations in CH_3_, CH_2_, and CH groups. Peaks at 1609 and 1584 cm^−1^ are connected with the skeletal C–C vibrations of the whole aromatic ring [43]. 

In the spectra of film containing estradiol prepared by both methods, the characteristic peaks of estradiol were remarkably changed. The peak from free O–H groups cannot be seen; only a broad band around 3400 cm^−1^ is observed, which could be due to overlap of peaks from the polymers and the drug. From Figure 7, the results of the FTIR spectra show the absence of most of the characteristic peaks of estradiol. The similarity between the blank and the drug-loaded film drug peak could be due to the high absorption peak of the film constituents. No significant IR spectra difference was observed in films prepared by the nano-emulsion and co-solvency methods.

### 3.13. Scanning Electron Microscopy (SEM)

The morphology and topographic appearance of buccal films containing estradiol were analysed with the help of SEM, as shown in Figure 8. The results show that pure estradiol appeared as irregular particles without well-defined shapes indicating the high crystalline nature of the drug [44].

The films prepared by co-solvency and nano-emulsion approaches showed continuous sheets with less rough surfaces compared to pure estradiol (Figure 8). This suggests that all formulation constituents were mixed and uniformly distributed in the film. The few pores and some rough surfaces that can be seen on the film surface formed by the co-solvency method could be due to rapid evaporation of ethanol during drying (Figure 8b). 

### 3.14. DSC Studies

Change in the physical state as well as the thermodynamic properties of the drug were evaluated using the DSC technique. Melting and solid-phase transformations are indicated by endothermic or exothermic peaks. The thermograms of estradiol, physical mixture, as well as blank and drug-loaded films prepared by both methods are shown in Figure 9. The melting point and crystalline nature of estradiol were indicated by a sharp endothermic peak observed at 179 °C [45]. However, films prepared by both co-solvency and nano-emulsion did not show the characteristic peak at 179 °C in the thermogram (Figure 9). This result illustrates the loss of crystallinity of estradiol and also confirms that the drug was molecularly dispersed in the film. Similarly, no characteristics peaks were noticed in the blank films. 

### 3.15. XRD Studies

XRD studies were used to examine the crystallinity of estradiol loaded into the film and also to confirm the results of the DSC studies. The diffractograms of films loaded with estradiol by both co-solvency and nano-emulsion are illustrated (Figure 10). The diffractograms of the pure drug and blank films as well as their corresponding physical mixtures were added for comparison. 

It was observed that the pure drug exhibited a diffraction pattern with numerous distinctive peaks indicating its highly crystalline state. The most abundant peaks were observed at 2θ values of 13.4°, 15.9°, and 18.4° and peaks of lower intensity 2θ values at 20.6°, 22.8°, and 26.8° [46].

The diffraction patterns of individual components for pure drug, physical mixture, blank, and drug-loaded films are shown in Figure 10. Drug-loaded film prepared by the nano-emulsion method showed no characteristic peaks indicating its amorphous state. Blank and drug-loaded films prepared by the co-solvency method show one broad peak at 2θ  =  20°, which could be due to HPMC [47]. The XRD profile of the physical mixture demonstrated the crystalline peaks for estradiol and other components corresponding to 2θ values of 9.8°, 13.0°, and 15.0°. This finding strongly suggests that the drug was distributed homogeneously in the polymeric matrix in an amorphous state. The XRD results of the drug-loaded films further support the DSC study and evidence that the drug is molecularly dispersed within the matrix.

### 3.16. Prediction of In Vivo Performance

The in vivo performance of nano-emulsion-based film was evaluated using a convolution technique in the R programming language. A convolution model is used to predict plasma drug concentrations based on in vitro release. The relationship between measured quantities (in vitro release and plasma drug concentrations) was modelled directly in a single stage using the convolve function in R software (the detailed script is included in Appendix A). Accordingly, the plasma concentration time profile of estradiol release from buccal film was predicted. Predicted blood levels along with the derived pharmacokinetic parameters (Tmax and Cmax) were assessed for a dose of 1.46 mg, for the films prepared by the nano-emulsion approach. The predicted Cmax and Tmax were found to be 740.74 ng mL^−^^1^ and 7 min, respectively (Figure 11). The predicted Cmax is higher than the previously reported plasma profile of estradiol using different routes, and the predicted Tmax is smaller compared to previous reports [11,17]. These predicted in vivo concentration time-curves give some insight into the relative clinical behaviour of the two films relative to intravenous administration and may help optimize the design of future in vivo studies of the buccal films. The broad conclusion for both films is that release is rapid and could produce higher plasma concentration compared to oral and previously reported oro-mucosal dosage forms.

## 4. Conclusions and Future Directions

Mucoadhesive buccal film of estradiol was successfully prepared using the solvent casting method. Two approaches, namely, co-solvency and nano-emulsion, were used to load the drug into the formulation. The results of the FTIR, DSC, and XRD confirm the encapsulation of the drug into the films by both approaches. The films exhibited excellent disintegration and improved mechanical properties. Furthermore, the surface morphology evaluation using SEM revealed a sheet with less rough surface and with no evidence of phase separation or creamy.

About 100% of the drug was released within 7.5 and 13 min from films prepared by the nano-emulsion and the co-solvency method. The permeability study illustrated that films prepared using the nano-emulsion strategy achieved about 15% permeation of the drug within 10 h, three times higher than reported for oral tablets. Whilst ex vivo permeation through buccal mucosa gives valuable information about the kinetics of the drug, important factors that could alter the kinetics such as blood flow and dilution by saliva were not taken into consideration in the ex vivo model. Hence, in vivo work that quantifies the amount of drug absorbed through buccal mucosa has to be conducted. Of note, studies have suggested that nanoparticles protect drugs against degradation and dilution in the saliva by adhering to the buccal mucosa supporting the potential use of nano-emulsion for hormonal therapy [48]. Moreover, estradiol showed high binding or association to buccal mucosa, meaning the drug could possibly undergo sustained permeation [49]. 

The predicted in vivo plasma drug concentration time profile indicates that films prepared by nano-emulsion resulted in higher concentration of estradiol in the blood compared to the previously reported oral and oromucosal administration confirming its potential use for treatment of menopausal symptoms and related complications [16].

Finally, the in vivo prediction model needs to be validated using in vivo data. 

## Figures and Tables

**Figure 1 pharmaceutics-14-00542-f001:**
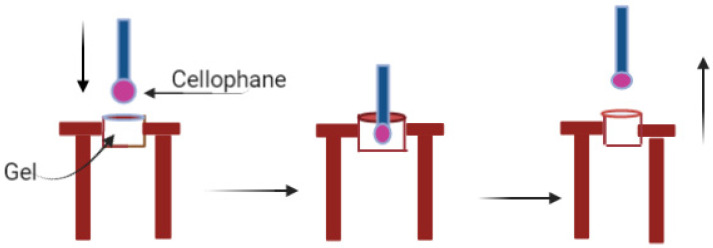
Schematic diagram of muco-adhesion test.

**Figure 2 pharmaceutics-14-00542-f002:**
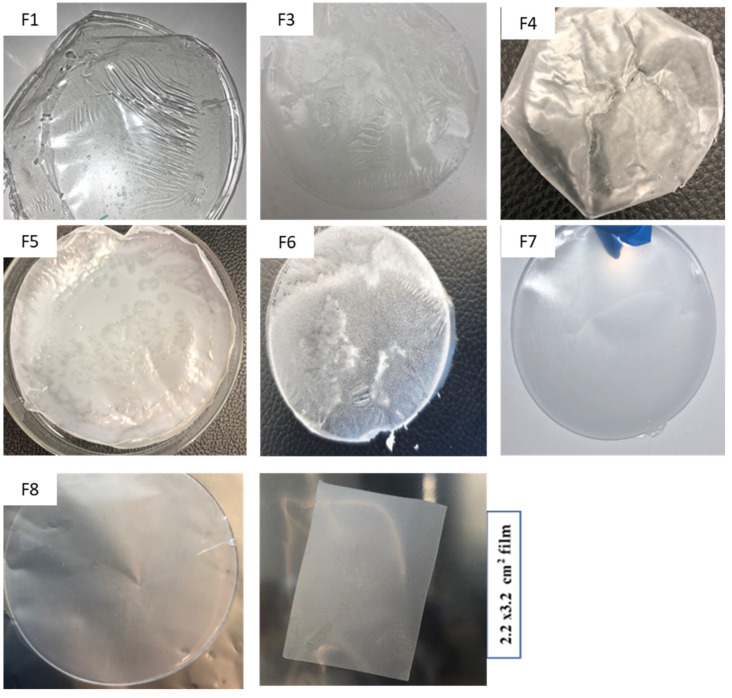
Physical appearance (digital photographs) of selected films prepared using different composition of polymers (note: Table 1) F1–8 (optimised formulations: Refer Table 1).

**Figure 3 pharmaceutics-14-00542-f003:**
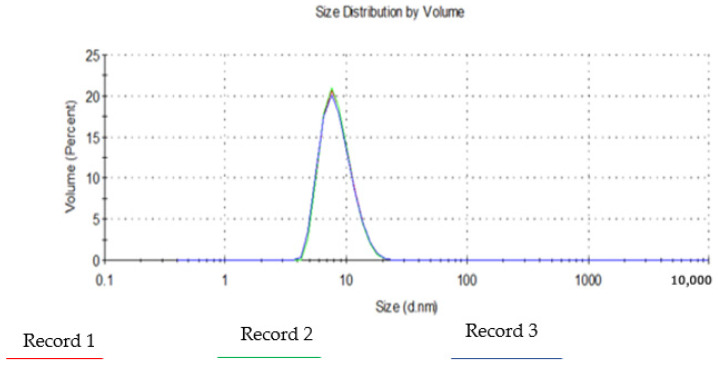
The average droplet size and PDI of nano-emulsion.

**Figure 4 pharmaceutics-14-00542-f004:**
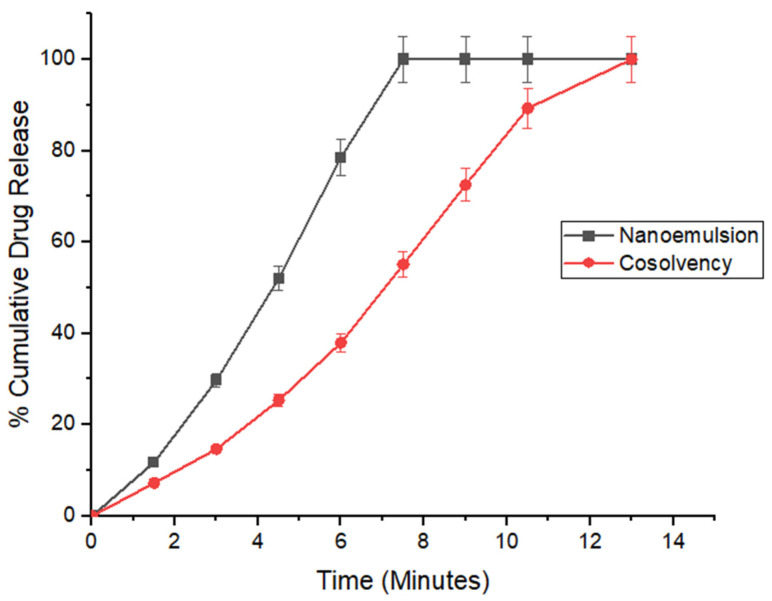
In vitro release kinetics of films prepared by co-solvency and nano-emulsion method.

**Figure 5 pharmaceutics-14-00542-f005:**
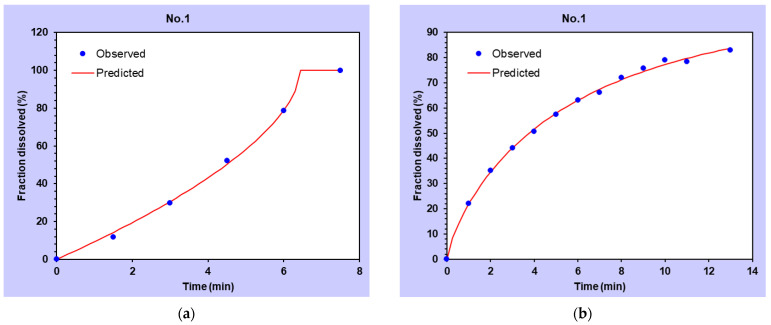
A release parameter fitted for estradiol films by (**a**) Hopfenberg and (**b**) Weibull for films prepared by nano-emulsion and co-solvency method.

**Figure 6 pharmaceutics-14-00542-f006:**
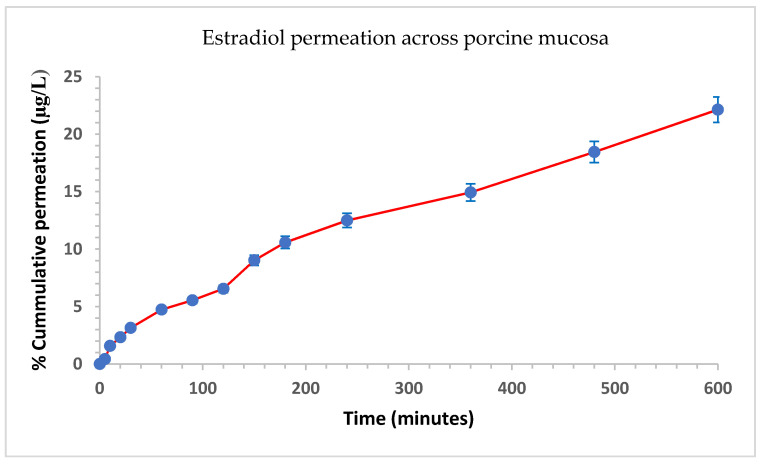
Permeation of estradiol across porcine mucosa from film prepared by nano-emulsion method.

**Figure 7 pharmaceutics-14-00542-f007:**
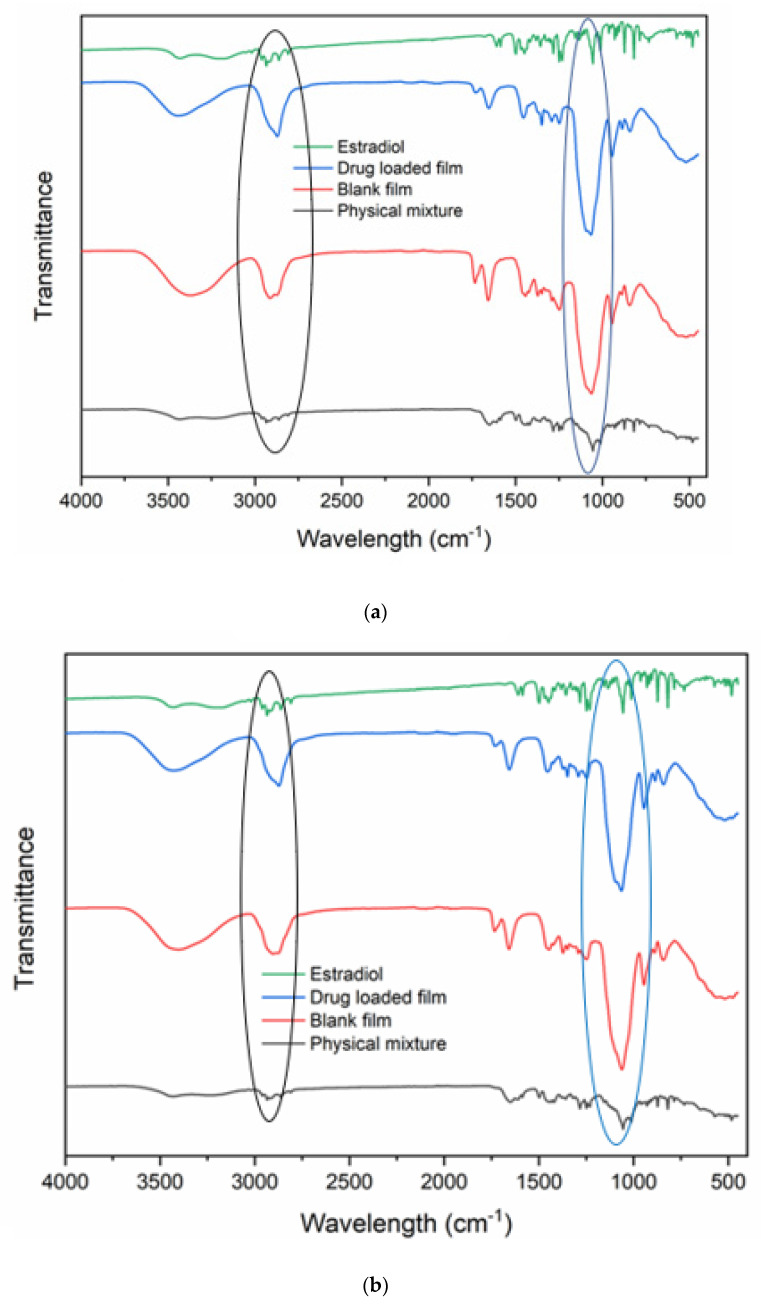
FTIR spectra of pure drug, physical mixture, blank, and drug-loaded films ((**a**) co-solvency; (**b**) nano-emulsion).

**Figure 8 pharmaceutics-14-00542-f008:**
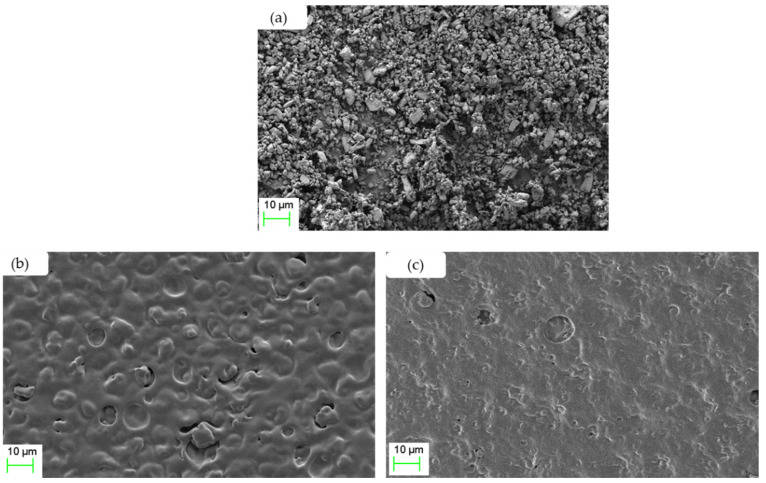
Representative scanning electron microscopy images of (**a**) pure estradiol, (**b**) the surface of the drug-loaded co-solvency film, and (**c**) drug-loaded nano-emulsion film.

**Figure 9 pharmaceutics-14-00542-f009:**
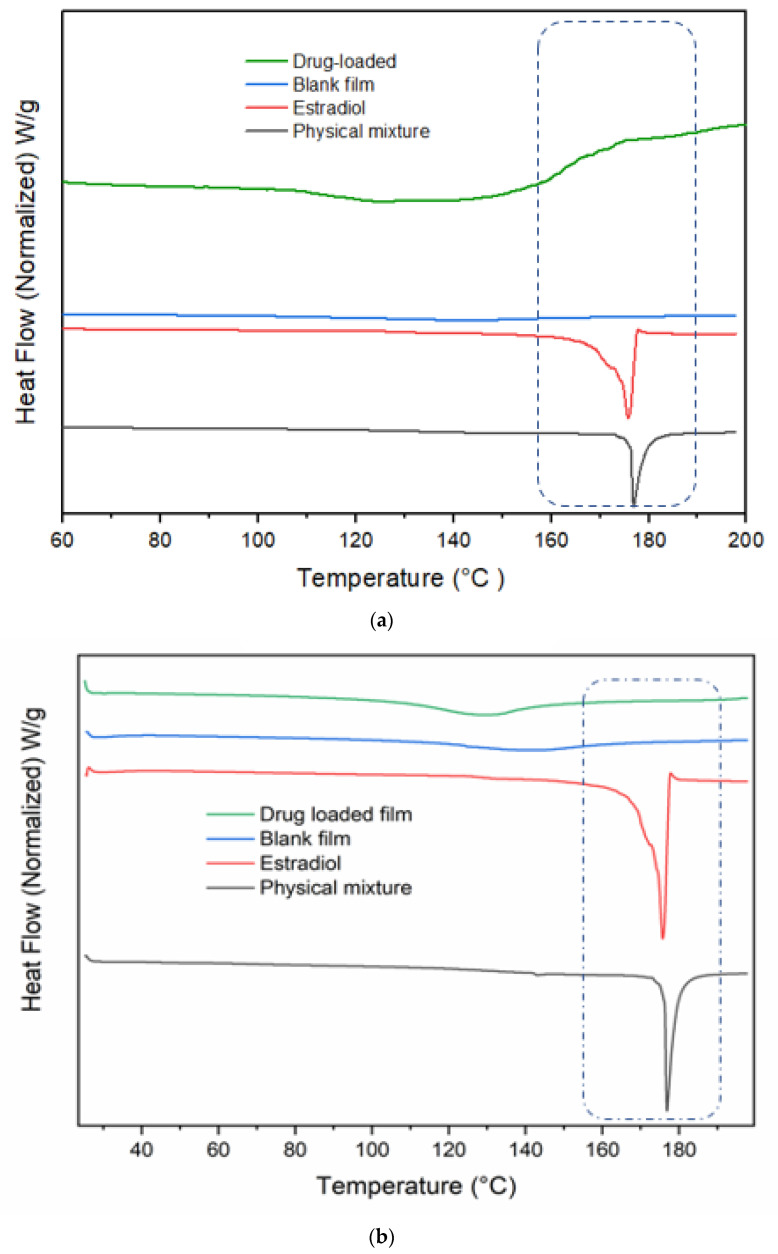
DSC thermographs of estradiol, equimolar physical mixture, blank and drug-loaded films ((**a**)co-solvency, (**b**) nano-emulsion).

**Figure 10 pharmaceutics-14-00542-f010:**
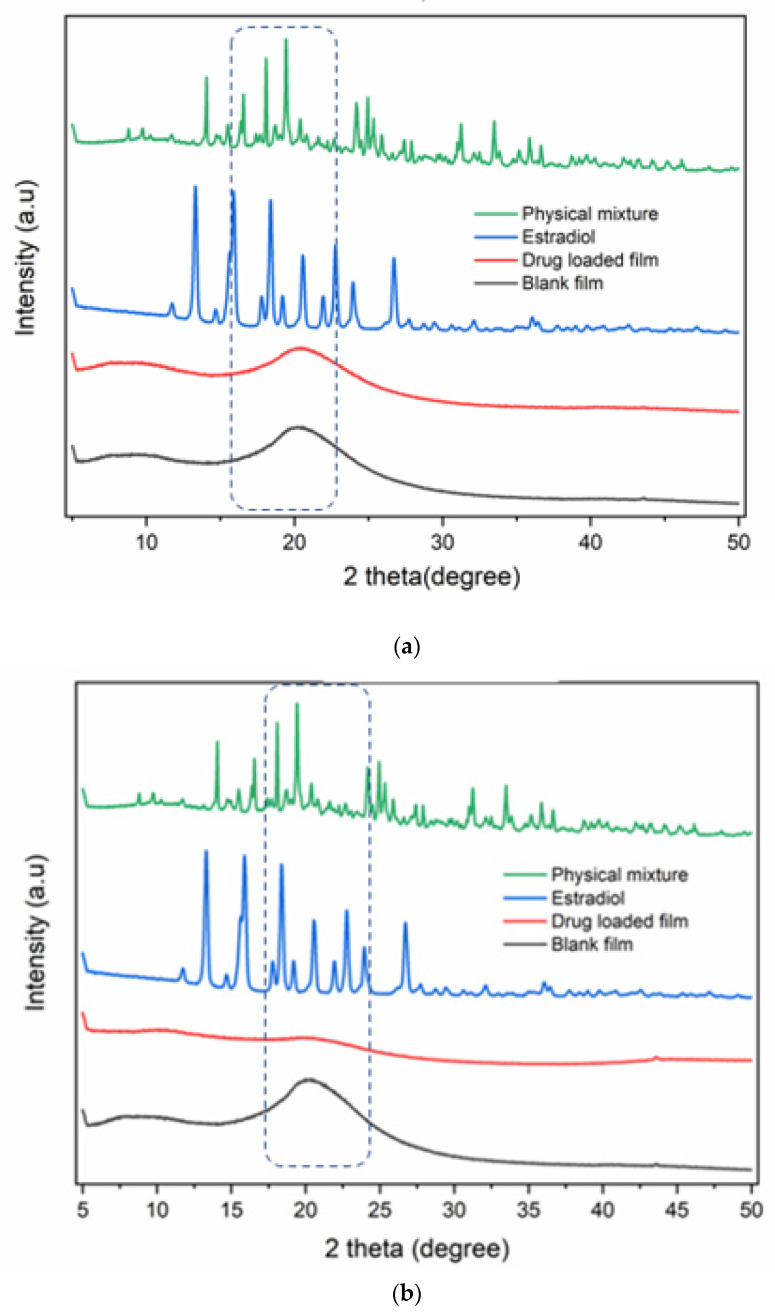
X-ray diffraction spectra of pure estradiol, blank, and drug-loaded film and their corresponding physical mixture ((**a**) co-solvency method, (**b**) nano-emulsion). The blue rectangle shows an area where characteristic peaks of the drug are observed.

**Figure 11 pharmaceutics-14-00542-f011:**
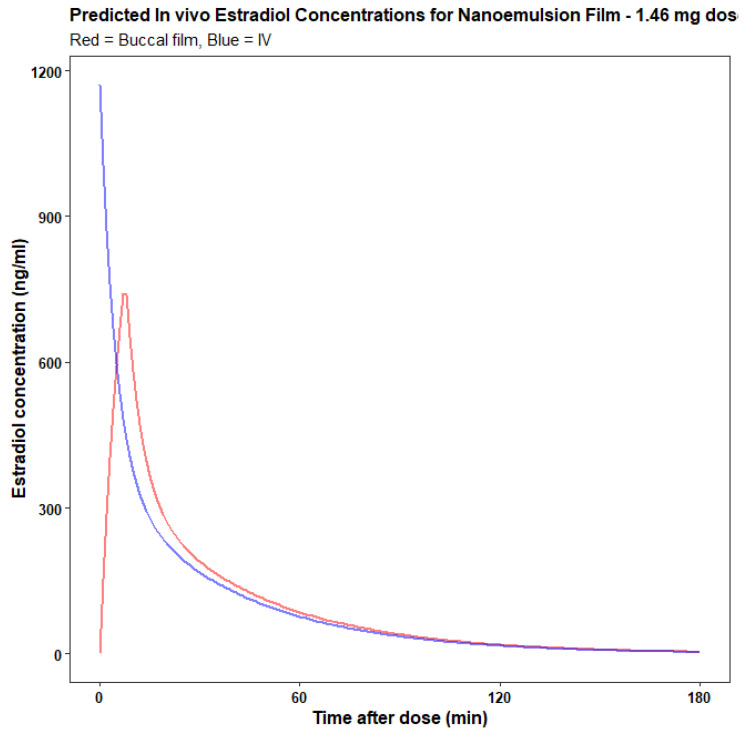
Predicted plasma-time profile of estradiol from films prepared by nano-emulsion approach.

**Table 1 pharmaceutics-14-00542-t001:** Optimized formulations to prepare estradiol-loaded mucoadhesive buccal films.

Ingredients	F1	F2	F3	F4	F5	F6	F7	F8
Estradiol (g)						0.08	0.08	0.08
PVA (g)	0.5	0.5	0.75			0.75	1	1
HPMC (g)		1	1			1	1.5	1.5
PVP (g)							0.4	0.4
Kollicoat(g)				4	4			
Xanthan (g)							0.015	0.015
Glycerin (mL)		0.2						
PEG (mL)	1		1.5		0.5	1.5	2.5	2.5
Tween 80 (mL)							0.5	0.5
Transcutol (mL)								1
Ethanol (mL)							12	
Citric acid (g)							0.125	0.125
Sucralose (g)							0.25	0.25
Water (QS) (mL)	50	50	50	50	50	50	50	50

F—formulation.

**Table 2 pharmaceutics-14-00542-t002:** Tensile strength and % elongation of films prepared by co-solvency and nano-emulsion method (*n* = 3).

	Co-Solvency	Nano-Emulsion	*p*-Value
	Mean ± SD	Mean ± SD	
Tensile strength (MPa)	0.51 ± 0.06	0.51 ± 0.01	0.811
Percent elongation	11.05 ± 0.52	12.3 ± 2.46	0.43

**Table 3 pharmaceutics-14-00542-t003:** Weight and thickness of mucoadhesive estradiol buccal films (3.2 cm × 2.2 cm) prepared by co-solvency and nano-emulsion approach (*n* = 3).

Film Preparation Approach	Weight (mg)		Thickness (mm)	
	Mean ± SD	*p*-Value	Mean ± SD	*p*-Value
Co-solvency	0.102 ± 0.004	<0.001	0.14 ± 0.01	0.492
Nano-emulsion	0.133 ± 0.00		0.15 ± 0.01	

**Table 4 pharmaceutics-14-00542-t004:** Droplet size and PDI of nanoparticles incorporated into estradiol buccal film.

Z-Average (d.nm)	14.92
**PdI**	0.487
**Result Quality**	Good

**Table 5 pharmaceutics-14-00542-t005:** Amount of estradiol loaded (mg) into a film prepared by co-solvency and nano-emulsion approaches (*n* = 3).

Film Preparation Approach	Mean ± SD	% Drug Recovery	*p*-Value
Co-solvency	1.47 ± 0.11	98.25%	0.971
Nano-emulsion	1.46 ± 0.27	97.50%	

**Table 6 pharmaceutics-14-00542-t006:** Release parameters of fitted experimental data for estradiol films prepared by cosolvency and nanoemulsion.

Model Name	Equation	Goodness of Fit Parameter	Cosolvency Film	Nanoemulision Film
		R2 adjusted	0.9689	0.9768
Zero order	F = k0 × t	RMSE	6.4343	5.926
		AIC	54.2245	33.0089
		R2 adjusted	0.8511	0.8596
First order	F = 100 × [1 − Exp(−k1 × t)]	RMSE	14.0805	14.5686
		AIC	68.3212	43.803
		R2 adjusted	0.7687	0.7953
Higuchi	F = kH × t0.5	RMSE	17.5484	17.5899
		AIC	72.2843	46.0645
		R2 adjusted	0.9837	0.9982
Korsmeyer-Peppasb	F = kkP × tn	RMSE	4.6546	1.6313
		AIC	49.1944	18.1902
		R2 adjusted	0.8924	0.8993
Hixson-Crowell	F = 100 × [1 − (1 − kHC × t)3]	RMSE	11.9675	12.3362
		AIC	65.3945	41.8071
		R2 adjusted	0.9933	0.9983
Hopfenberg	F = 100 × [1 − (1 − kHB × t)n]	RMSE	2.9814	1.6108
		AIC	41.1765	18.0388
		R2 adjusted	0.7007	0.7312
Baker-Lonsdale	3/2 × [1 − (1 − F/100)2/3] − F/100 = kBL × t	RMSE	19.9617	20.1554
		AIC	74.6036	47.6983
		R2 adjusted	0.9841	0.9983
Peppas-Sahlinc	F = k1 × tm + k2 × t2m	RMSE	4.6033	1.604
		AIC	49.6076	18.2614
		R2 adjusted	0.9972	0.9912
Weibull	F = 100 × {1 − Exp[− ((t − Ti)β)/α]}	RMSE	1.9443	3.6431
		AIC	34.0936	28.1058

**Notes:** F, Percentage of drug released at time t; *k*0, Zero order release constant; *k*1, First order release constant; *k*H, Higuchi release constant; *kk*P, Release rate constant, and bn, diffusional release exponent; *k*HC, Release constant relevant to Hixson–Crowell model; *k*HB, Combined constant corresponding to Hopfenberg model in which *k*HB = *k*0/(C0 × α0) where *k*0, erosion rate constant, C0, initial drug concentration in the matrix, α0, initial radius for a slab/cylinder/sphere structure, and n, 1, 2, and 3 for the slab, cylinder, and sphere structure, respectively; *k*BL, Combined constant related to Baker–Lonsdale model in which *k*BL = [3 × D × Cs/(r02 × C0)] where D, diffusion coefficient, Cs, saturation solubility, r0, initial radius for a sphere/cylinder/slab structure, and C0, initial drug concentration in the matrix; ck1, Constant relevant to the Fickian kinetics, and ck2, constant relevant to Case-II relaxation kinetics, and cm, diffusional release exponent; Abbreviations: R2 adjusted, adjusted coefficient of determination; RMSE, Root mean squared error; AIC, Akaike Information Criterion.

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
