# Peer review of "Mucoadhesive Buccal Film of Estradiol for Hormonal Replacement Therapy: Development and In-Vivo Performance Prediction"

_pharmaceutics, 2022, doi:10.3390/pharmaceutics14030542_

Round 1

Reviewer 1 Report

A very interesting research article on estradiol mucoadhesive buccal film formulations. The introduction part despite the fact that is very long is very informative and well written. Methods are well explained but lack of references. Results and Discussion are well presented. Some points to consider:

Lines 20, 121, etc: check english grammar

Lines 381, 394 : text editing

Line 557: Figure 8b: time (d) means days?

Reviewer 2 Report

Transbuccal drug delivery could be a tempting alternative to traditional oral and parenteral drug delivery, thus the manuscript "Mucoadhesive Buccal Film of Estradiol for Hormonal Replacement Therapy: Development and In-vivo performance prediction" is relevant and novel. However, several issues must be addressed before its publication:

  1. The conclusions are dubious and should be written more precisely and clearly. Although the experiments carried out in different model systems (in vitro film disintegration and in vitro drug release to artificial saliva solution, and drug permeation across porcine buccal mucosa in Franz diffusion cells) were successful when considered separately, the overall result of transbuccal estradiol delivery seems to be unsuccessful. While the dissolution of the film prepared by nano-emulsion approach was complete within 30s and all the drug was released within 7.5 min, it took 10 h of film exposure to buccal mucosa to achieve only 15% of transbuccal drug delivery. This, under real conditions, by 10th hour neither drug-loaded film nor drug itself will remain in the oral cavity and thus even this small percentage of drug transbuccal delivery will hardly be achieved. By this time, the whole drug dose will be swallowed resulting in all the adverse effects cited by the authors for the traditional oral estadiol administration. Thus, the part of the Conclusion in lines 624-632 describing the advantage of the obtained film over other reported oral and oro-mucosal forms is incorrect and should be improved.
  2. The information on toxicity/non-toxicity of film components should be added.
  3. Lines 20-21: Please modify the sentence "Although oral estrogens are the most widely used forms of oral estrogens... " to underline the difference between various ways of oral estrogen administartion.
  4. Please use only one form of the hormone name in the text: oestradiol or estradiol.
